# A Diffusion-Model of Joint Interactive Navigation

**Matthew Niedoba**[1,2]    **J. Wilder Lavington**[1,2]    **Yunpeng Liu**[1,2]    **Vasileios Lioutas**[1,2]

**Justice Sefas**[1,2]    **Xiaoxuan Liang**[1,2]    **Dylan Green**[1,2]    **Setareh Dabiri**[2]

**Berend Zwartsenberg**[2]    **Adam Scibior**[1,2]    **Frank Wood**[1,2]

[1] University of British Columbia, [2] Inverted AI

mniedoba@cs.ubc.ca

## Abstract

Simulation of autonomous vehicle systems requires that simulated traffic participants exhibit diverse and realistic behaviors. The use of prerecorded real-world traffic scenarios in simulation ensures realism but the rarity of safety critical events makes large scale collection of driving scenarios expensive. In this paper, we present DJINN – a diffusion based method of generating traffic scenarios. Our approach jointly diffuses the trajectories of all agents, conditioned on a flexible set of state observations from the past, present, or future. On popular trajectory forecasting datasets, we report state of the art performance on joint trajectory metrics. In addition, we demonstrate how DJINN flexibly enables direct test-time sampling from a variety of valuable conditional distributions including goal-based sampling, behavior-class sampling, and scenario editing.

## 1  Introduction

Accurate simulations are critical to the development of autonomous vehicles (AVs) because they facilitate the safe testing of complex driving systems [15]. One of the most popular methods of simulation is virtual replay [46], in which the performance of autonomous systems are evaluated by replaying previously recorded traffic scenarios. Although virtual replay is a valuable tool for AV testing, recording diverse scenarios is expensive and time consuming, as safety-critical traffic behaviors are rare [17]. Methods for producing synthetic traffic scenarios of specific driving behaviors are therefore essential to accelerate AV development and simulation quality.

Producing these synthetic traffic scenarios involves generating the joint future motion of all the agents in a scene, a task which is closely related to the problem of trajectory forecasting. Due to the complexity of learning a fully autonomous end-to-end vehicle controller, researchers often opt to split the problem into three main tasks [52]: perception, trajectory forecasting, and planning. In trajectory forecasting, the future positions of all agents are predicted up to a specified future time based on the agent histories and the road information. Due to the utility of trajectory forecasting models in autonomous vehicle systems along with the availability of standard datasets and benchmarks to measure progress [4, 53], a variety of effective trajectory forecasting methods are now available. Unfortunately, most methods produce *deterministic* sets of trajectory forecasts *per-agent* [47, 9] which are difficult to combine to produce realistic joint traffic scenes [30].

Generative models of driving behavior have been proposed as an alternative to deterministic trajectory forecasting methods for traffic scene generation [40, 46]. These models re-frame trajectory forecasting as modeling the joint distribution of future agent state conditioned on past observations and map context. However, given that the distribution of traffic scenes in motion forecasting datasets are similar to real-world driving, modelling the data distribution does not ensure that models will generate rare, safety critical events.

To alleviate these issues we propose DJINN, a model which generatively produces joint traffic scenarios with *flexible conditioning*. DJINN is a diffusion model over the joint states of all agents in the scene. Similar to [30], our model is conditioned on a flexible set of agent states. By modifying the conditioning set at test-time, DJINN is able to draw traffic scenarios from a variety of conditional distributions of interest. These distributions include sampling scenes conditioned on specific goal states or upsampling trajectories from sparse waypoints. Additionally, the joint diffusion structure of DJINN enables test-time diffusion guidance. Utilizing these methods enables further control over the conditioning of traffic scenes based on behavior modes, agent states, or scene editing.

We evaluate the quality of sampled trajectories with both joint and ego-only motion forecasting on the Argoverse [4] and INTERACTION [53] datasets. We report excellent ego-only motion forecasting and outperform Scene Transformer on joint motion forecasting metrics. We further demonstrate both DJINN's flexibility and compatibility with various forms of test-time diffusion guidance by generating goal-directed samples, examples of cut-in driving behaviors, and editing replay logs.

## 2   Related Work

**Trajectory Forecasting:** A wide variety of methods have been proposed to address the problem of trajectory forecasting. Two attributes which divide this area of work are the output representation type and the agents for which predictions are made. The most common class of models deterministically predict the distribution of ego agent trajectories using a weighted trajectory set either with or without uncertainties. Due to the applicability of this representation as the input for real-time self-driving planners, there are numerous prior methods of this type. Some approaches rasterize the scene into a birdview image and use CNNs to predict a discrete set of future trajectories for the ego agent [5, 3, 32]. The convolutional architecture of these methods captures local information around the agent well, but the birdview image size and resolution limit the ability to capture high speed and long-range interactions. To address these challenges, other prior approaches encode agent states directly either by using RNNs [47, 39, 27], polyline encoders [7, 9] or 1D convolutions [24]. Agent features can be combined with roadgraph information in a variety of ways including graph convolutional networks [24, 1] or attention [29, 27].

To control the distribution of predicted trajectories, several methods have utilized mode or goal conditioning. One approach is to directly predict several goal targets before regressing trajectories to those targets [54, 9, 51, 8]. An alternate approach is to condition on trajectory prototypes [3] or latent embeddings [47].

Predicting joint traffic scenes using per-agent marginal trajectory sets is challenging due to the exponential growth of trajectory combinations. Recent approaches aim to rectify this by producing joint weighted sets of trajectories for all agents in a scene. M2I [45] generates joint trajectory sets by producing "reactor" trajectories which are conditioned on marginal "influencer" trajectories. Scene Transformer [30], which uses a similar backbone architecture to our method, uses a transformer [48] network to jointly produce trajectory sets for all agents in the scene.

As an alternative to deterministic predictions, multiple methods propose generative models of agent trajectories. A variety of generative model classes have been employed including Normalizing Flows [36], GANs [10, 38] or CVRNNs [40, 46]. Joint generative behavior models can either produce entire scenarios in one shot [10, 38, 36], or produce scenarios by autoregressively "rolling-out" agent trajectories [40, 46].

**Diffusion Models:** Diffusion models, proposed by Sohl-Dickstein et al. [41] and improved by Ho et al. [12] are a class of generative models which approximate the data distribution by reversing a forward process which gradually adds noise to the data. The schedule of noise addition can be discrete process or represented by a continuous time differential equation [44, 21].We utilize the diffusion parameterization introduced in EDM [21] in our work for its excellent performance and separation of training and sampling procedures.

This class of models have shown excellent sample quality in a variety of domains including images [12, 6], video [11, 14] and audio [22]. In addition, diffusion models can be adapted at test-time through various conditioning mechanisms. Classifier [6] and classifier-free guidance [13] have enabled powerful conditional generative models such as text conditional image models [34, 37] while editing techniques [26, 31] have enabled iterative refinement of generated samples.

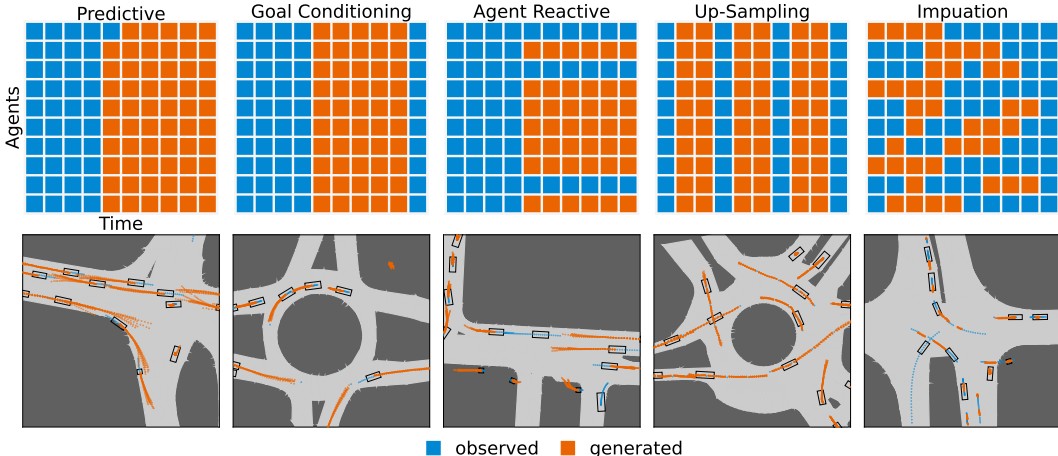

Figure 1: **Top:** Five example observation masks $\mathcal{O}$ demonstrating potential conditioning inputs to DJINN. Each element of each mask corresponds to the boolean value of $\mathcal{O}$ for that agent state. Individual agents are shown in rows, with timesteps as columns. **Bottom:** Generated traffic scenes corresponding to the type of observation masks above.

One recent application of diffusion models is planning. Diffuser [20] uses diffusion models to generate trajectories for offline reinforcement learning tasks. They condition their samples using classifier guidance to achieve high rewards and satisfy constraints. Trace and Pace [35] utilizes diffusion planning for guided pedestrian motion planning. In the vehicle planning domain, Controllable Traffic Generation (CTG) [55] builds on Diffuser, using diffusion models to generate trajectories which satisfy road rule constraints. Like CTG, our method also models the future trajectories of road users using diffusion models. However, our approach differs from CTG in terms of output and our methods of conditioning. In CTG, marginal per-agent trajectory samples are combined into a joint scene representation by "rolling-out" a portion of each agent's trajectory before drawing new samples per-agent. By contrast, DJINN models the full joint distribution of agent trajectories in one shot, with no re-planning or roll-outs required. The authors of CTG condition their model exclusively on the past states of other agents and the map, and use classifier-guidance to condition their samples to follow road rules. In our method, we demonstrate conditioning on scene semantics via classifier guidance *as well as* conditioning on arbitrary state observations, including the past or future states of each agent, and control the strength of conditioning using classifier-free guidance as demonstrated in Fig. 1.

## 3 Background

### 3.1 Problem Formulation

Our work considers traffic scenarios consisting of $A$ agents across $T$ discrete timesteps driving on a roadway described by a set of roadgraph features $\mathbf{M}$. Each agent $a \in \{1 \ldots, A\}$ in the scene at time $t \in \{1, \ldots T\}$ is represented by a state $\mathbf{s}_t^a = \{x_t^a, y_t^a, \theta_t^a\}$ consisting of its 2D position $(x_t, y_t)$ and heading $\theta_t$. The joint representation of the scene $\mathbf{x}$ is the combination of all agents across all timesteps $\mathbf{x} = \{s_t^a | a \in \{1, \ldots, A\}, t \in \{1, \ldots, T\}\} \in \mathbb{R}^{A \times T \times 3}$. We assume scenes are distributed according to an unknown distribution $p_{data}(\mathbf{x})$.

We introduce a model which is conditioned on the map features $\mathbf{M}$ and can moreover be flexibly conditioned on arbitrary set of observed agent states. For the latter purpose, we consider a boolean variable $\mathcal{O} \in \{0, 1\}^{A \times T}$. We denote that a state in the scene is observed if $\mathcal{O}_t^a = 1$. Using $\mathcal{O}$, we partition the scene into two components. The observed portion of the scene is defined as $\mathbf{x}_{obs} = \{\mathbf{s}_t^a | \mathbf{s}_t^a \in \mathbf{x}, \mathcal{O}_t^a = 1\}$ while the unobserved, latent portion is $\mathbf{x}_{lat} = \mathbf{x} \setminus \mathbf{x}_{obs}$. Figure 1 shows five choices for $\mathcal{O}$ and their corresponding tasks. Our ultimate goal is to learn a conditional distribution over the set of all latent agent states $\mathbf{x}_{lat}$ given the observed states $\mathbf{x}_{obs}$ and the map $\mathbf{M}$, by modelling $p(\mathbf{x}_{lat} | \mathbf{x}_{obs}, \mathbf{M})$. Using this probabilistic framework, we can represent conditional

distributions corresponding to various trajectory forecasting tasks by modifying the observation mask $\mathcal{O}$ and the corresponding conditioning set $\mathbf{x}_{obs}$.

## 3.2 Diffusion Models

Diffusion models [41, 12] are a powerful class of generative models built upon a diffusion process which iteratively adds noise to the data. In the continuous time formulation of this process [44, 21], this iterative addition is described by a stochastic differential equation (SDE)

$$d\mathbf{x}_\tau = \mu(\mathbf{x}_\tau, \tau)d\tau - \sigma(\tau)d\mathbf{w}. \tag{1}$$

Here, $\tau \in [0, \tau_{max}]$ where $\tau_{max}$ is a fixed, large constant, $\mu(\mathbf{x}_\tau, \tau)$ is the drift function and $\sigma(\tau)$ is the diffusion coefficient which scales standard Brownian motion $\mathbf{w}$. Note that our work has two notions of time. Throughout we will use $t$ to denote the "scenario time" and $\tau$ to represent "diffusion time". We express the marginal distribution of $\mathbf{x}_\tau$ at diffusion time $\tau$ as $p(\mathbf{x}_\tau)$, with $p(\mathbf{x}_0)$ corresponding to the data distribution $p_{data}(\mathbf{x})$. Typically, $\mu(\mathbf{x}_\tau, \tau)$, $\sigma(\tau)$, and $\tau_{max}$ are chosen such the conditional density $p(\mathbf{x}_\tau | \mathbf{x}_0)$ is available in closed form and that $p(\mathbf{x}_{\tau_{max}})$ approximates a tractable Gaussian distribution $\pi(\mathbf{x})$. Notably, for every diffusion SDE, there exists a corresponding probability flow (PF) ordinary differential equation (ODE) [44] whose marginal probability densities $p(\mathbf{x}_\tau)$ match the densities of Eq. (1)

$$d\mathbf{x}_\tau = \left[ \mu(\mathbf{x}_\tau, \tau) - \frac{1}{2}\sigma(\tau)^2 \nabla_x \log p(\mathbf{x}_\tau) \right] d\tau. \tag{2}$$

Using the PF ODE, samples are generated from a diffusion model by integrating Eq. (2) from $\tau = \tau_{max}$ to $\tau = 0$ with initial condition $\mathbf{x}_{\tau_{max}} \sim \pi(\mathbf{x}_{\tau_{max}})$ using an ODE solver. Typically integration is stopped at some small value $\epsilon$ for numerical stability. Solving this initial value problem requires evaluation of the *score function* $\nabla_{\mathbf{x}_\tau} \log p(\mathbf{x}_\tau)$. Since $p(\mathbf{x}_\tau)$ is not known in closed form, diffusion models learn an approximation of the score function $\mathbf{s}_\theta(\mathbf{x}_\tau, \tau) \approx \nabla_{\mathbf{x}_\tau} \log p(\mathbf{x}_\tau)$ via score matching [16, 43, 44].

A useful property of diffusion models is the ability to model conditional distributions $p(\mathbf{x}_0 | y)$ at test-time using guidance. Given some conditional information $y$, the key idea of guidance is to replace the score function in the PF ODE with an approximate *conditional* score function $\nabla_{\mathbf{x}_\tau} \log p(\mathbf{x}_\tau | y)$.

By using the gradient of a pretrained classifier $p_\phi(y | \mathbf{x}_\tau)$, glassifier guidance [6] approximates the conditional score function through the a linear combination of the unconditional score function and the classifier gradient. The parameter $\alpha$ controls the strength of the guidance

$$\nabla_{\mathbf{x}_\tau} \log p(\mathbf{x}_\tau | y) \approx \mathbf{s}_\theta(\mathbf{x}_\tau, \tau) + \alpha \nabla_{\mathbf{x}_\tau} \log p_\phi(y | \mathbf{x}_\tau). \tag{3}$$

One major drawback of classifier guidance is the need to train an external classifier. Instead, classifier-free guidance [13], utilizes a conditional score network $\mathbf{s}_\theta(\mathbf{x}_\tau, \tau, y)$. Then, a weighted average of the conditional and unconditional scores is used to estimate the conditional score function.

$$\nabla_{\mathbf{x}_\tau} \log p(\mathbf{x}_\tau | y) \approx \lambda \mathbf{s}_\theta(\mathbf{x}_\tau, \tau, y) + (1 - \lambda)\mathbf{s}_\theta(\mathbf{x}_\tau, \tau). \tag{4}$$

Here $\lambda$ is a scalar parameter which controls the strength of the guidance. In both cases, the approximate conditional score can be substituted into Eq. (2) to draw conditional samples from $p(\mathbf{x}_0 | y)$.

## 4 DJINN

Our approach models the joint distribution agent states $p(\mathbf{x}_{lat} | \mathbf{x}_{obs}, \mathbf{M})$ conditioned on a set of observed states and the map context. For this purpose, we employ a diffusion model which diffuses directly over $\mathbf{x}_{lat}$ – the unobserved states of each agent in the scene for $t = \{1, \ldots T\}$. An important aspect of our method is the choice of observation mask $\mathcal{O}$ and observation set $\mathbf{x}_{obs}$ on which we condition. For this purpose we introduce a distribution over observation masks $p(\mathcal{O})$ which controls the tasks on which we train our model.

In the design of our diffusion process, we follow the choices from EDM [21], setting $\mu(\mathbf{x}_{lat,\tau}, \tau) = \mathbf{0}$ and $\sigma(\tau) = \sqrt{2\tau}$ from Eq. (2). We also utilize their score function parameterization

$$\nabla_{\mathbf{x}_{lat,\tau}} \log p(\mathbf{x}_{lat,\tau}|\mathbf{x}_{obs}, \mathbf{M}, \mathbf{c}) = \frac{D_\theta\left(\mathbf{x}_{lat,\tau}, \mathbf{x}_{obs}, \mathbf{M}, \mathbf{c}, \tau\right) - \mathbf{x}_{lat,\tau}}{\tau^2}. \tag{5}$$

Here $D_\theta$ is a neural network which approximates the latent portion of the noise free data $\mathbf{x}_{lat,0}$. In addition to $\mathbf{x}_{lat,\tau}$ and $\tau$, in our work $D_\theta$ also receives the map context $\mathbf{M}$, the clean observed states $\mathbf{x}_{obs}$ and $c$, a collection of unmodelled agent features per observed agent timestep such as velocity, vehicle size, or agent type. We train our network on a modification of the objective from EDM [21]

$$\mathbb{E}_{\mathbf{x}_0,\tau,\mathcal{O},\mathbf{x}_{lat,\tau}} \|D_\theta\left(\mathbf{x}_{lat,\tau}, \mathbf{x}_{obs}, \mathbf{M}, \mathbf{c}, \tau\right) - \mathbf{x}_{lat,0}\|_2^2. \tag{6}$$

Here, $\mathbf{x}_0 \sim p_{data}(\mathbf{x})$, $\mathbf{x}_\tau \sim p(\mathbf{x}_\tau|\mathbf{x}_0) = \mathcal{N}(\mathbf{x}, \tau^2\mathbf{I})$ and $\mathcal{O} \sim p(\mathcal{O})$. We compute our loss over $\tau \sim p_{train}$ – a log normal distribution which controls the variance of the noise added to the data. We set the mean and variance of $p_{train}$ according to [21].

We use the Heun 2nd order sampler from [21] to sample traffic scenarios with no changes to the reported hyperparameters. Empirically, we found that deterministic sampling, corresponding to integrating the PF ODE, leads to higher quality samples than using an SDE solver. Unless otherwise noted all samples are produced using 50 iterations of the ODE solver, which produces the highest quality samples as measured by ego and joint minADE and minFDE.

**Input Representation** An important choice for trajectory forecasting models is the reference frame for the agent states. In our work, the diffused agent states and observations $\mathbf{x}_{obs}$ are centered around an "ego agent," which is often specified in trajectory forecasting datasets as the primary agent of interest. We transform $\mathbf{x}_0$ such that the scene is centered on the last observed position of this arbitrary "ego agent" and rotated so the last observed heading of the ego agent is zero. We scale the positions and headings of all agents in each ego-transformed scene to a standard deviation of $0.5$.

We represent the map $\mathbf{M}$ as an unordered collection of polylines representing the center of each lane. Polylines are comprised of a fixed number of 2D points. We split longer polylines split into multiple segments and pad shorter polylines padded to the fixed length. Each point has a boolean variable indicating whether the element is padding. Polyline points are represented in the same reference frame as the agent states and are scaled by the same amount as the agent position features.

**Model Architecture** Our score estimator network $D_\theta$ is parameterized by a transformer-based architecture similar to [30]. The network operates on a fixed $[A, T, F]$ shaped feature tensor composed of one $F$ dimensional feature vector per agent timestep. We use sinusoidal positional embeddings [48] to produce initial feature tensors. Noisy and observed agent states $\mathbf{x}_\tau$, $\mathbf{x}_{obs}$, the time indices $t = \{1, \ldots, T\}$, and diffusion step $\tau$ are all embedded into $F$ dimensional embeddings. $\mathbf{x}_{lat,\tau}$ and $\mathbf{x}_{obs}$ are padded with zeros for observed and latent states respectively prior to embedding. A shared MLP projects the concatenated positional embeddings into a $F$ dimensional vector for each agent.

The main trunk of the network is comprised of a series of transformer layers [48]. Attention between all pairs of feature vectors is factorized into alternating time and agent transformer layers. In time transformer layers, self-attention is performed per-agent across each timestep of that agent's trajectory, allowing for temporal consistency along a trajectory. In agent transformer layers, self-attention is computed across all agents at a given time, updating each agent's features with information about the other agents at that time. We encode the map information $\mathbf{M}$ with a shared MLP that consumes flattened per-point and per-lane features to produce a fixed size embedding per lane. Cross attention between the collection of lane embeddings and agent states incorporates map information into the agent state features. Our network is comprised of 15 total transformer layers with a fixed feature dimension of 256. We use an MLP decoder after the final transformer layer to produce our estimate of $\mathbf{x}_{lat,0}$. A full representation of our architecture is available in Appendix A.

## 5  Guidance for Conditional Scene Generation

So far, we have outlined our method for generating joint traffic scenes using DJINN. Next, we describe how the diffusion nature of DJINN enables fine-grained control over the generation and modification of driving scenarios.

Table 1: Ego-only motion forecasting performance on Argoverse and INTERACTION datasets. minADE and minFDE metrics on both datasets indicate that DJINN produces ego samples which closely match the distribution of ego agent trajectories.

<table>
<tr><td colspan="3" align="center">(a) Argoverse test set</td></tr>
<tr><th>Method</th><th>minADE$_6$</th><th>minFDE$_6$</th></tr>
<tr><td>Jean [27]</td><td>0.98</td><td>1.42</td></tr>
<tr><td>mmTransformer [25]</td><td>0.87</td><td>1.34</td></tr>
<tr><td>DenseTNT [9]</td><td>0.88</td><td>1.28</td></tr>
<tr><td>MultiPath++[47]</td><td>0.79</td><td>1.214</td></tr>
<tr><td>DCMS [50]</td><td>0.77</td><td>1.14</td></tr>
<tr><td>SceneTransformer [30]</td><td>0.80</td><td>1.23</td></tr>
<tr><td>Ours</td><td>1.02</td><td>1.65</td></tr>
</table>

<table>
<tr><td colspan="3" align="center">(b) INTERACTION validation set</td></tr>
<tr><th>Method</th><th>minADE$_6$</th><th>minFDE$_6$</th></tr>
<tr><td>DESIRE [23]</td><td>0.32</td><td>0.88</td></tr>
<tr><td>TNT [54]</td><td>0.21</td><td>0.67</td></tr>
<tr><td>ReCoG [28]</td><td>0.19</td><td>0.65</td></tr>
<tr><td>ITRA [40]</td><td>0.17</td><td>0.49</td></tr>
<tr><td>StarNet [19]</td><td>0.13</td><td>0.38</td></tr>
<tr><td>SAN [18]</td><td>0.10</td><td>0.29</td></tr>
<tr><td>Ours</td><td>0.14</td><td>0.39</td></tr>
</table>

Table 2: Ego-only and joint metrics comparing DJINN to a jointly trained Scene Transformer model on the Argoverse validation set. DJINN produces better joint samples than SceneTransformer when measured by minSceneADE and minSceneFDE.

| Method | minADE$_6$ | minFDE$_6$ | minSceneADE$_6$ | minSceneFDE$_6$ |
|---|---|---|---|---|
| Scene Transformer (Joint) | **0.848** | **1.398** | 1.019 | 1.835 |
| Ours | 0.871 | 1.409 | **0.895** | **1.758** |

## 5.1 Classifier-free Guidance

In Scene Transformer [30], a masked sequence modelling framework is introduced for goal-directed and agent-reactive scene predictions. One limitation of this approach is that conditioning is performed on precise agent states while future agent states or goals are usually uncertain. We mitigate this limitation through the use of classifier-free guidance.

We assume access to a set of precise observations $\mathbf{x}_{obs}$, and some set of additional agent states $\mathbf{x}_{cond}$ on which we wish to condition our sample. For instance, $\mathbf{x}_{cond}$ may include agent goals upon which we wish to condition. Let $\mathbf{x}'_{obs} = \{\mathbf{x}_{obs} \cup \mathbf{x}_{cond}\}$. Based on Eq. (4), the conditional score is through a weighted average of the score estimate conditioned on $\mathbf{x}_{obs}$ and the estimated conditioned on $\mathbf{x}'_{obs}$

$$
\begin{aligned}
\nabla_{\mathbf{x}_{lat,\tau}} \log p(\mathbf{x}_{lat,\tau}|\mathbf{x}'_{obs}) \approx &\lambda \frac{D_\theta\left(\mathbf{x}_{lat,\tau}, \mathbf{x}'_{obs}, \mathbf{M}, \mathbf{c}, \tau\right) - \mathbf{x}_{lat,\tau}}{\tau^2} \\
&+ (1-\lambda) \frac{D_\theta\left(\mathbf{x}_{lat,\tau}, \mathbf{x}_{obs}, \mathbf{M}, \mathbf{c}, \tau\right) - \mathbf{x}_{lat,\tau}}{\tau^2}.
\end{aligned}
\tag{7}
$$

To facilitate classifier-free conditioning, we train DJINN on a $p(\mathcal{O})$ representing varied conditioning tasks. These tasks include conditioning on agent history, agent goals, windows of agent states, and random agent states. A full overview of our task distribution is given in Appendix B.

## 5.2 Classifier Guidance

Many driving behaviors of individual or multiple agents can be categorized by a class $y$ based on their geometry, inter-agent interactions or map context. Examples of classes include driving maneuvers such as left turns, multi agent behaviors such as yielding to another agent, or constraints such as trajectories which follow the speed limit. DJINN uses classifier guidance to conditioned scenes on these behavior classes. Given a set of example scenes corresponding to a behavior class $y$, we train a classifier to model $p_\phi(y|\mathbf{x})$. Using Eq. (3) we approximate the conditional score for conditional sampling. Importantly, due to the joint nature of our representation, classifiers for per-agent, multi-agent or whole-scene behaviors can be all used to condition sampled traffic scenes.

## 5.3 Scenario Editing

One benefit of sampling traffic scenes at once instead of autoregressively is the ability to edit generated or recorded traffic scenarios through stochastic differential editing [26]. Given a traffic scene $\mathbf{x}$, a user can manually modify the trajectories in the scene to produce a "guide" scene $\mathbf{x}'$ which approximates

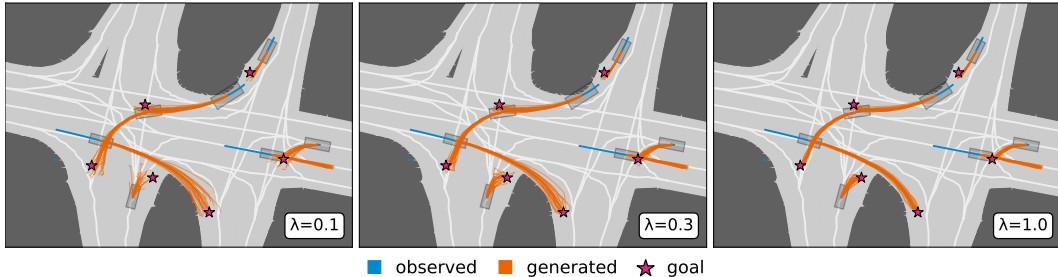

Figure 2: The effect of classifier-free guidance weight on the spread of trajectories for goal conditioned sampling. Samples drawn from the INTERACTION validation set conditioned using classifier-free guidance on a goal state (star). As the guidance weight increases, deviation from the goals decreases.

the desired trajectories in the scene. The guide scene is used to condition the start of a truncated reverse diffusion process by sampling $\mathbf{x}_{\tau_{edit}} \sim \mathcal{N}(\mathbf{x}', \tau_{edit}\mathbf{I})$ where $\tau_{edit}$ is an intermediate time in the diffusion process between $0$ and $\tau_{max}$. Then, the edited scene is produced by integrating the PF ODE using the same ODE solver, starting from initial condition $\tau_{edit}$. Through the stochastic differential editing, the guide scene is translated into a realistic traffic scene with agent trajectories which approximate the guide trajectories. We empirically find $\tau_{edit} = 0.8$ to be a good trade-off between generating realistic trajectory scenes and maintaining the information of the guide scene.

## 6 Experiments

### 6.1 Motion Forecasting Performance

To measure the quality of the samples from DJINN, we evaluate our method on two popular motion prediction datasets, matching $\mathcal{O}$ during training to match each dataset. For the INTERACTION dataset [53] scenes, we observe the state of all agents over the first second of the scene and generate the next three seconds. On the Argoverse dataset [4] our model observes agent states over the first two seconds of the scene and generates the next three seconds. Training hyperparameters for both models are found in Appendix A.

We note that both INTERACTION and Argoverse metrics measure an ego-only trajectory-set using minADE and minFDE over 6 trajectories. Since DJINN produces stochastic samples of entire traffic scenes, a set of 6 random trajectories may not cover all future trajectory modes. To alleviate this, we draw a collection of 60 samples for each scenario and fit a 6 component Gaussian mixture model with diagonal covariances using EM in a method similar to [47]. We use the means of the mixture components as the final DJINN prediction for motion forecasting benchmarks.

We present DJINN's performance on motion forecasting in Table 1 with Argoverse results in Table 1a and INTERACTION results in Table 1b. On INTERACTION, DJINN generates excellent ego vehicle trajectories, with similar minFDE and minADE to state of the art methods on this dataset. On the Argoverse test set we produce competitive metrics, although our results lag slightly behind top motion forecasting methods. We hypothesize that our lower performance on Argoverse is due to the lower quality agent tracks in this dataset when compared to INTERACTION.

We further analyze the *joint* motion forecasting performance of DJINN. To this end, we measure the Scene minADE and minFDE proposed by [2] which measures joint motion forecasting performance over a collection of traffic scenes. We compare DJINN against a reproduction of Scene Transformer trained for joint motion forecasting, using their reported hyperparameters. Ego-only and Scene motion forecasting performance is shown in Table 2. Although Scene Transformer predicts slightly better ego vehicle trajectories, we demonstrate DJINN has superior joint motion forecasting capabilities.

### 6.2 State-conditioned Traffic Scene Generation

While DJINN is able to draw samples for motion forecasting benchmarks by conditioning on past observations of the scene, a key benefit of our approach is the ability to flexibly condition at test-time

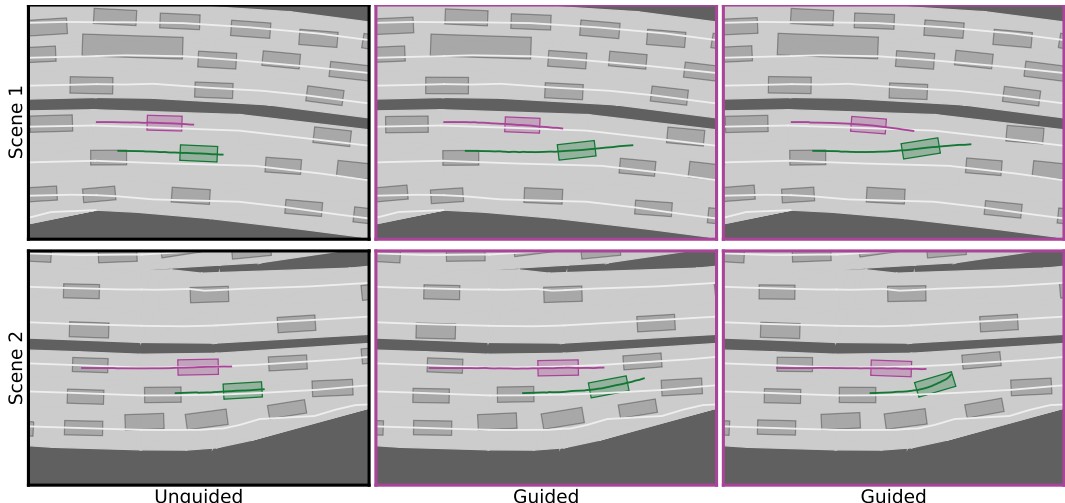

Figure 3: Examples of synthetic cut-in behaviors generated using classifier guidance. Samples are generated from the INTERACTION validation set conditioned on the first 10 agent states. Applying classifier guidance causes the other agent (green) to cut in front of the ego agent (purple). We generate trajectories for all agents in the scene, but other agent trajectories have been omitted for clarity

based on arbitrary agent states. We illustrate this test-time conditioning in Fig. 1 by generating samples from five conditional distributions which correspond to use-cases for our model.

Specifying exact agent states on which to condition can be challenging. One approach is to utilize the states of a prerecorded trajectory to produce conditioning inputs. However, if one wishes to generate a trajectory which deviates from a recorded trajectory, there is uncertainty about the exact states on which to condition. In Fig. 2, we demonstrate how classifier-free guidance can be utilized to handle user uncertainty in conditioning agent states. In this example, we set the observation set $\mathbf{x}_{obs}$ to the first ten states of each agent's recorded trajectory. Further, we create a conditional observation set $\mathbf{x}'_{obs}$ by augmenting $\mathbf{x}_{obs}$ with a goal state for each agent drawn from a normal distribution centered on the ground-truth final position of each agent, with 1m variance. We sample traffic scenes with varying levels of classifier-free guidance strength, drawing two conclusions. First, DJINN is robust to goals which do not match the recorded final agent states. Secondly, the strength of the classifier guidance weight controls the emphasis of the goal conditioning, resulting in trajectory samples which cluster more tightly around the specified goal as the guidance strength is increased. With low guidance weight, the samples are diverse, and do not closely match the specified goal position. As the weight increases, the spread of the trajectory distribution tightens, especially for fast, longer trajectories. These properties give users finer control over the distribution of traffic scenes when there is uncertainty over the conditioning states.

### 6.3 Conditional Generation from Behavior Classes

We now continue to demonstrate the flexibility of our approach by considering test-time conditioning of our model on specific driving behaviors through classifier guidance. Specifically, we highlight the ability to condition DJINN on the behavior class of cut-in trajectories by conditioning our INTERACTION trained model with a cut-in classifier.

A "cut-in" occurs when one vehicle merges into the path of another, often requiring intervention by the cut-off driver. We selected this behavior to demonstrate how classifier guidance can be used with our joint representation to sample scenes conditioned on the behavior of multiple agents. We condition DJINN trained on INTERACTION using a simple cut-in classifier. To train the classifier, we first mined a dataset of cut-in behaviors trajectory pairs from the "DR_CHN_Merging_ZS" location – a highway driving scene with some cut-in examples. Each trajectory pair is comprised of an "ego" and an "other" agent. We define a positive cut-in as a case where the future state of the other agent at time $t_{other}$ overlaps with a future state of the ego agent at time $t_{ego}$ such that $t_{ego} - 3s < t_{other} < t_{ego}$. Further, we filter cases where the initial state of the other agent overlaps with any part of the ego

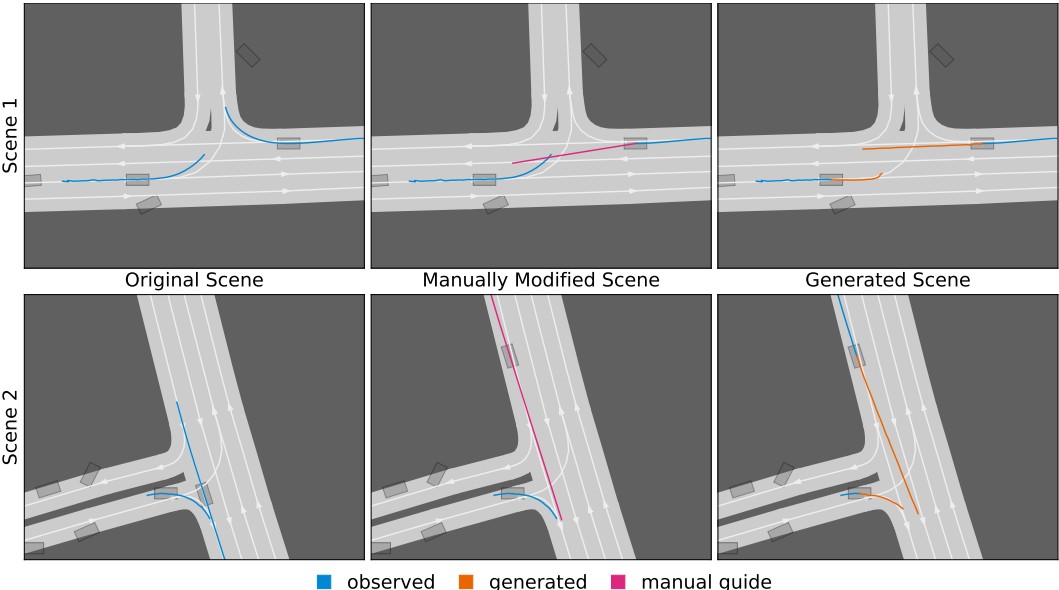

Scene 1

Original Scene · Manually Modified Scene · Generated Scene

Scene 2

■ observed ■ generated ■ manual guide

Figure 4: Two scenario fine-tuning examples (one per row) based on Argoverse validation set scenarios. **Left**: original scene with ground-truth trajectories shown for two interacting vehicles, vehicle positions at the same time index for all agents. **Middle**: a manual edit of one agent's trajectory in each scene. One (top) replaces a right turn with a forward continuation, the other (bottom) shifts a trajectory back in space to cause a complex interaction to occur near the end of the trajectory. **Right**: the resulting stochastic differential edit of the original scenario. Both rows of the last column illustrate joint reactivity to the new trajectories arising from the edit; in the top row the left-turning vehicle yields and in the bottom row both trajectories shift to avoid collision.

trajectory to eliminate lane following cases. We label a negative cut-in case as any other pair of trajectories in which the minimum distance between any pair of ego and other states is less than 5m.

Using these heuristics, we collect a dataset of 2013 positive and 296751 negative examples. We trained a two layer MLP classifier with 128 dimensions per hidden layer. The classifier takes as input the diffused trajectories of each agent, the validity of each timestep and the diffusion time $\tau$. Using this classifier, we generate synthetic cut-in scenarios via Eq. (3). Examples of our synthetic cut-in scenarios are found in Fig. 3. The generated scenarios clearly demonstrate our model can be conditioned to create synthetic cut-in behaviors. These synthetic examples provide evidence that given a collection of trajectories exemplifying a behavior mode, or a heuristic which can be used to generate example trajectories, DJINN can be conditioned to generate synthetic examples representing that behavior mode. This finding further expands the flexibility of our model to generate trajectory samples from valuable conditional distributions.

### 6.4 Scenario Fine-Tuning

We exhibit another method of controlling the traffic scenarios generated with DJINN through fine-tuning. Since DJINN diffuses entire traffic scenes without iterative replanning, we are able to use stochastic differential editing to modify the sampled scenes. Given a recorded or sampled traffic scene, differential stochastic editing can be used to fine-tune the scene through the use of a manually specified guide. In Fig. 4, we demonstrate how DJINN can fine-tune existing scenarios to produce new scenarios with realistic trajectories but complex interactions. Using two recorded validation set scenes from Argoverse, we aim to edit the scenes to generate more interactive trajectories between the agents. For this purpose, we generate an guide scene $\mathbf{x}_{guide}$ by manually adjusting the trajectories in each scene so that the future paths of two of the agents will intersect. Through stochastic differential editing, we show that DJINN is able to produce realistic driving scenes which shift the guide scene trajectories to maintain their interactivity but avoid collisions between agents.

# 7 Conclusions

In this work, we present DJINN – a diffusion model of joint traffic scenes. By diffusing in a joint agent state representation, DJINN can be adapted at test time to a variety of modeling tasks through guidance methods and scenario editing. The power of this scenario generation model opens exciting possibilities. Future research may expand the variety of guidance classifiers such as utilizing the classifiers proposed in [55] for traffic-rule constraint satisfaction. Another promising avenue of research is scaling DJINN for faster scenario generation. Although flexible, the diffusion structure of DJINN makes scenario generation relatively slow due to the iterative estimation of the score function. Distillation techniques such as consistency models [42] may be helpful in this regard to improve the number of score estimates required per sample. Future work may also consider scaling the length and agent count in generated scenarios to improve the complexity of behaviors which can be generated. Other areas of future work include using DJINN in a model predictive control setting (hinted at in the predictive mask of Fig. 1) in which an ego action is scored using statistics of ego-action conditioned joint trajectories from DJINN.

## Acknowledgements

We acknowledge the support of the Natural Sciences and Engineering Research Council of Canada (NSERC), the Canada CIFAR AI Chairs Program, Inverted AI, MITACS, the Department of Energy through Lawrence Berkeley National Laboratory, and Google. This research was enabled in part by technical support and computational resources provided by the Digital Research Alliance of Canada Compute Canada (alliancecan.ca), the Advanced Research Computing at the University of British Columbia (arc.ubc.ca), Amazon, and Oracle.

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

# Appendix

## A  Model Details

### A.1  Preconditioning

Following EDM [21], we precondition $D_\theta$ by combining $\mathbf{x}_{lat,\tau}$ and the output of our network $F_\theta$ using scaling factors

$$D_\theta(\mathbf{x}_{lat,\tau}, \mathbf{x}_{obs}, \mathbf{M}, \mathbf{c}, \tau) = c_{skip}(\tau)\mathbf{x}_{lat,\tau} + c_{out}(\tau)F_\theta(c_{in}\mathbf{x}_{lat,\tau}, \mathbf{x}_{obs}, \mathbf{M}, \mathbf{c}, c_{noise}(\tau)). \quad (8)$$

We use the scaling values reported in [21] without modification but report them in Table 3 for convenience.

Table 3: Scaling Functions for Preconditioning

| Scaling Factor | Function |
|---|---|
| $c_{skip}$ | $\sigma_{data}^2/(\tau^2 + \sigma_{data}^2)$ |
| $c_{out}$ | $\tau \cdot \sigma_{data}/\sqrt{\tau^2 + \sigma_{data}^2}$ |
| $c_{in}$ | $1/\sqrt{\tau^2 + \sigma_{data}^2}$ |
| $c_{noise}$ | $\frac{1}{4}\ln(\tau)$ |

Here, $\sigma_{data}$ is the standard deviation of the diffusion features. We scale the positions and headings of the agent so that $\sigma_{data}$ is $0.5$ for all diffusion features.

### A.2  Architecture

DJINN utilizes a transformer based architecture for $F_\theta$. An overview of the model structure is shown in Fig. 5.

**Feature Encoding**

We encode the observed agent states $\mathbf{x}_{obs}$, the noisy latent states $\mathbf{x}_{lat,\tau}$, the scenario time $t$ and the diffusion time $\tau$ using sinusoidal positional encoding [48]. We represent the scenario time as an integer index increasing from 0 to $T$ with 0 corresponding to the earliest agent states. For each of the encoded features, we produce a 256-dimensional encoding vector. An important hyperparameter for sinusoidal positional embeddings are the maximum and minimum encoding periods which we report in Table 4.

Table 4: Maximum and minimum positional encoding periods for DJINN input features

| Feature | Minimum Period | Maximum Period |
|---|---|---|
| $\mathbf{x}_{obs}, \mathbf{x}_{lat,\tau}$ | 0.01 | 10 |
| $t$ | 1 | 100 |
| $\tau$ | 0.1 | 10,000 |

The concatenation of the positional encodings with additional agent state features $\mathbf{c}$ are fed through an MLP to form the input to the main transformer network. The additional agent state features consist of the agent velocity, the observed mask, and the agent size which is available for INTERACTION only. The input MLP is shared across all agent states and contains two linear layers with hidden dimension 256 and ReLU non-linearities.

**Roadgraph Encoding**

DJINN is conditioned on the geometry of the roadgraph through a collection of lane center polylines. Each polyline is comprised of an ordered series of 2D points which represent the approximate center of each driving lane. We fix the length of each polyline to 10 points. We split polylines longer than this threshold into approximately equal segments, and pad shorter polylines with zeros. We utilize a boolean feature to indicate which polyline points are padded. Unlike [30], we do not use a PointNet [33] to encode the roadgraph polylines. Instead, we encode the polylines into a 256-dimensional vector per polyline using a simple MLP. To generate the input to this MLP, we concatenate the position

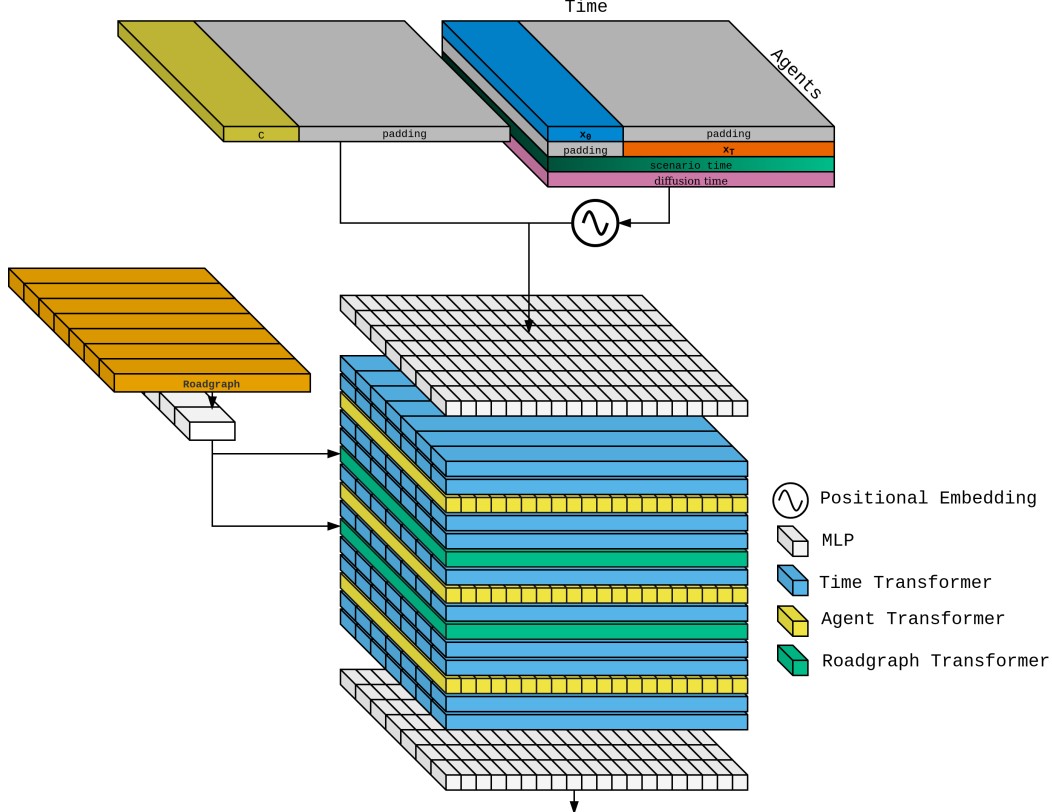

Figure 5: An overview of the DJINN architecture. The main network structure is comprised of time, agent and roadgraph attention layers. Features are encoded using positional encodings and MLPs. The output of the network is an estimate of the de-noised agent states.

and padding mask for each polyline, along with any additional per-polyline features present in the dataset. For both datasets, the MLP is comprised of four linear layers, with a hidden dimensionality of 256 and ReLU non-linearities.

**Transformer Network**

The main transformer backbone of DJINN is comprised of 15 transformer layers which perform self-attention over the time and agent dimension, and cross attention with the roadgraph encodings. We utilize the same transformer layers as those proposed in [30], but modify the number of layers and their ordering. Specifically, include more time transformer layers as we found this produced smoother trajectories. All attention layers consume and produce 256 dimensional features per-agent state. We use four heads for each attention operation, and a 1024 dimensional hidden state for in the feed forward network. In the transformer layers, we use the pre-layernorm structure described in [49].

Due to batching and agents which are not tracked for the duration of the traffic scene, there is padding present in the agent feature tensor. The transformer layers account for padding in the scene by modifying the attention masks so that padded agent states are not attended to.

**Output MLP**

We use a two layer MLP with hidden dimension 256 to produce the final output for $F_\theta$. We produce a three dimensional vector per agent state for INTERACITON and two-dimensional vector for Argoverse since headings are not provided in the dataset.

### A.3 Training details

We train DJINN on two A100 GPUs for 150 epochs. We utilize the Adam optimizer with learning rate of 3E-4 and default values for $\beta_1$ and $\beta_2$. We use a linear learning rate ramp up, scaling from 0 to 3E-4 over 0.1 epochs. We set the batch size to 32. We clip gradients to a maximum norm of 5. Training takes approximately 6 days to complete from scratch.

## B   Observation Distribution

We train DJINN over a variety of observation masks $\mathcal{O}$ by randomly drawing masks from a training distribution $p(\mathcal{O})$. Table 5 outlines this training task distribution. We refer to the length of agent state history observation for each dataset as $t_{obs}$ and the total number of timesteps as $T$. $\mathcal{U}$ indicates a uniform distribution over integers.

Table 5: Task distribution for training DJINN. The training observation mask $\mathcal{O}$ is sampled from this distribution with probabilities given in the rightmost column.

| Task | Description | Probability |
|------|-------------|-------------|
| Predictive | Observe states where $t \in [0, t_{obs}]$. | 50% |
| Goal-Conditioned | Observe states where $t \in [0, t_{obs}]$ and the final state of 3 random agents. | 25% |
| Agent-Conditioned | Observe states where $t \in [0, t_{obs}]$ and the entire trajectory of 3 random agents. | 10% |
| Ego-Conditioned | Observe states where $t \in [0, t_{obs}]$ and the entire ego-agent trajectory. | 10% |
| Windowed | Observe states where $t \in [0, t_{start}]$ and $t \in (t_{start} + t_{obs}, T]$ where $t_{start} \sim \mathcal{U}(0, t_{obs})$. | 5% |
| Upsampling | Observe every $t_{obs}/T$ states, starting from $t_{start} \sim \mathcal{U}(0, t_{obs}/T)$. | 5% |
| Imputation | Randomly sample observing each state with probability $t_{obs}/T$. | 5% |

## C   Additional Qualitative Results

Fig. 6 shows additional qualitative samples from DJINN on the INTERACTION dataset for a subset of the observation masks outlined in Fig. 1. Each row in the figure corresponds to a different tasks, abd each element of the row is a sampled traffic scene.

## D   Additional Quantitative Results

### D.1   Effect of Observation Distribution

To enable test-time conditioning through classifier-free guidance as outlined in section 6.2, we train DJINN on the observation distribution described in Appendix B. To quantify the effect of training on this distribution, we compare the sample quality of a DJINN model trained trained on the full observation distribution to one which is trained exclusively on the "Predictive Task." Table 6 shows the impact of the observation distribution as measured by trajectory forecasting metrics on samples drawn from INTERACTION dataset scenes.

Table 6 demonstrates that training on the full mixture of observation masks somewhat reduces the predictive performance of DJINN when compared to the model trained exclusively on the predictive task. However, the diversity of trajectories measured using MFD [40] increases when training on the more diverse distribution.

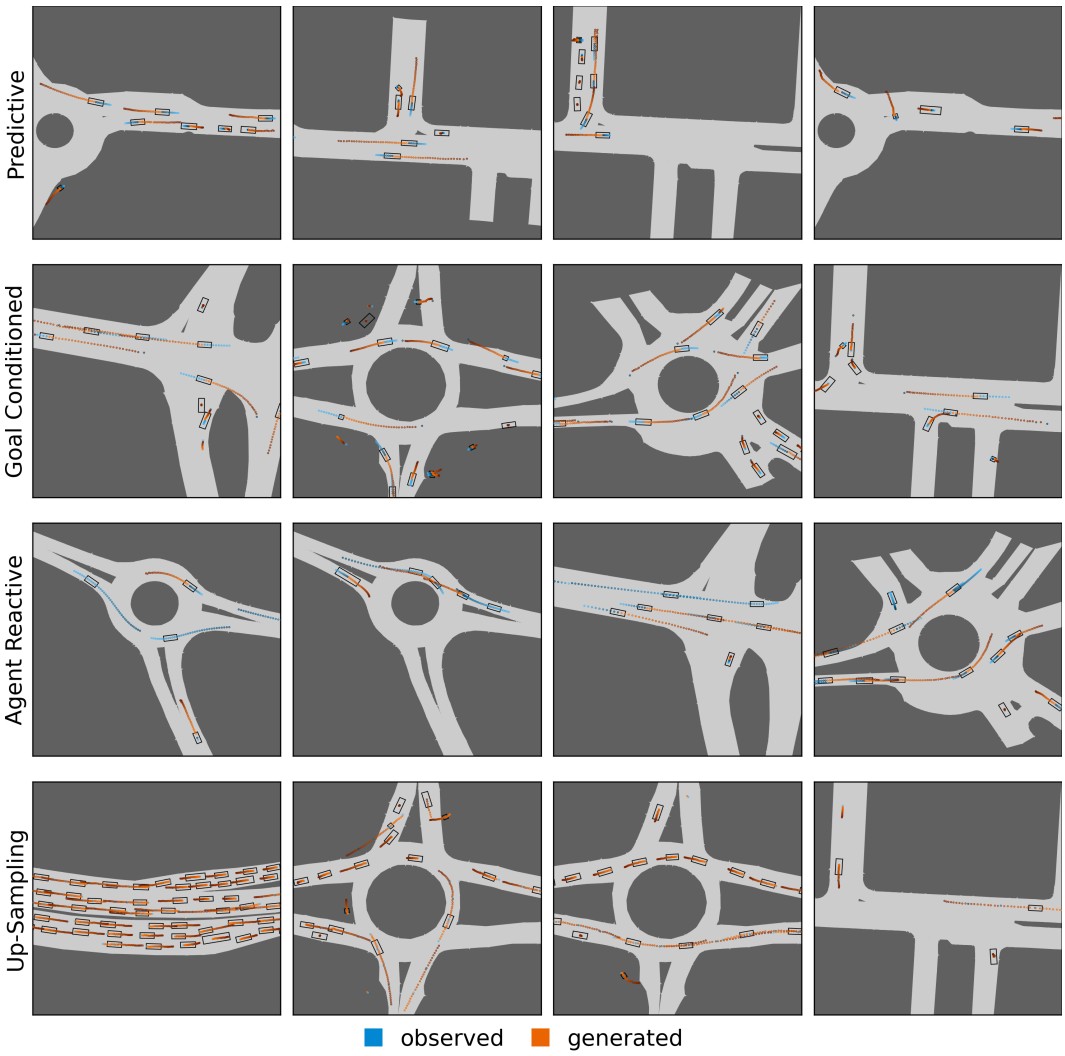

Figure 6: Additional generated traffic scenes from the INTERACTION validation set. Each row demonstrates samples generated using a different observation mask.

Table 6: Comparison of trajectory forecasting performance for models trained with varying observation distributions. Trajectory metrics are measured using 6 samples per scene on the INTERACTION validation set.

| Observations | minADE | minFDE | Scene minADE | Scene minFDE | MFD |
|---|---|---|---|---|---|
| Predictive | 0.21 | 0.49 | 0.35 | 0.91 | 2.33 |
| Mixture | 0.26 | 0.63 | 0.45 | 1.17 | 3.11 |

## D.2 Effect of Reduced Sampling Steps

The continuous time training procedure of DJINN enables test-time variation in the number of sampling steps. Table 7 outlines the effect of reducing the number of sampling steps from the 50 steps which are used in all other experiments.

Table 7 shows that reducing the number of sampling steps results in modest trajectory forecasting performance reductions up to 20 sampling steps across all metrics. Using 10 steps severely impacts the quality of sampled scenes across all metrics. As sampling time scales linearly with the number of sampling steps, reducing the number of sampling steps allows for a performance runtime tradeoff.

Table 7: Trajectory forecasting performance versus the number of timesteps used in the diffusion sampling procedure. Trajectory forecasting performance is measured using 6 samples per scene on the INTERACTION validation set.

| Diffusion Steps | minADE | minFDE | Scene minADE | Scene minFDE |
|---|---|---|---|---|
| 10 | 0.28 | 0.64 | 0.45 | 1.135 |
| 20 | 0.22 | 0.51 | 0.37 | 0.95 |
| 30 | 0.22 | 0.50 | 0.36 | 0.92 |
| 40 | 0.21 | 0.50 | 0.35 | 0.92 |
| 50 | 0.21 | 0.49 | 0.35 | 0.92 |

## D.3    Model Runtime

We compare DJINN's runtime to SceneTransformer [30], varying the input size as measured by the number of agents in the scene. Runtimes are measured across 1000 samples on a GeForce RTX 2070 Mobile GPU.

Table 8: Average scenario generation time for DJINN and Scene Transformer across varying scene sizes.

| Agent Count | Scene Transformer | DJINN - 50 Steps | DJINN - 25 Steps |
|---|---|---|---|
| 8 | 0.0126s | 0.574s | 1.15s |
| 16 | 0.0140s | 0.611s | 1.24s |
| 32 | 0.017s | 0.844s | 1.69s |
| 64 | 0.026s | 1.40s | 2.89s |

