# OpenReview forum: "A Diffusion-Model of Joint Interactive Navigation"
_NeurIPS.cc/2023/Conference — NeurIPS 2023 poster_

### Official Review · Reviewer_wJNr · 2023-07-06

**Soundness:** 3 good
**Presentation:** 3 good
**Contribution:** 3 good
**Rating:** 7
**Confidence:** 4

**Summary:**

The paper deals with the problem of generating vehicle trajectories at the scene level conditioned on a map and some known observations (e.g. past or future states). They propose a diffusion model based on SceneTransformer that is trained with observation conditionings with random masking to reflect multiple desired downstream applications including prediction, goal conditioning, and imputation.  Experiments show reasonable results on trajectory forecasting on both Argoverse and INTERACTION datasets, and demonstrate the flexibility of the model to several tasks including generating scenes based on high-level concepts (cut-in) and scenario editing.

**Strengths:**

While no technical component of DJINN is particularly novel, the paper presents a nice comprehensive exploration of using diffusion for traffic scenario modeling. This includes all different types of masking/conditioning, classifier-free guidance, classifier guidance, and scene editing with SDEdit.

Sec 3.2 is a succinct, yet informative, introduction to diffusion which will be good even for the unfamiliar reader in the driving community. In general, the paper is relatively self-contained, well-written, and easy to follow.

The conditioning approach makes the model very flexible to many different kinds of observations that enable use in several important downstream applications.

The paper goes beyond showing the usual motion forecasting applications (although it shows reasonable results on this too). The use of classifier guidance in Sec 6.3 to allow creating scenarios with more abstract specifications is a cool proof-of-concept that could be really useful for AV testing. Similarly for the scene editing application in Sec 6.4.

**Weaknesses:**

In some cases, it looks like conditioning the model on input observations is not strong enough to be consistent with the observations. E.g. Fig 1 bottom row for Goal Conditioning, Up-Sampling, and Imputation the generated parts of trajectories in orange do not always align well with observations in blue, i.e. there are jumps or discontinuities. It may be necessary to use test-time guidance (as in CTG [52] and [Trace and Pace, Rempe et al., CVPR 2023]) to better enforce consistency. Also, it would be good to have an evaluation to quantify this inconsistency. E.g., for up-sampling and imputation the generated portions could be compared to ground truth, or for goal conditioning how close the vehicle gets to the goal.

Some details of the masking procedure to allow this conditioning were also not clear. Does the model only output predictions for timesteps which there were no observations? Or does it generate a full trajectory and only the unmasked portion is visualized in the figures? If the latter, is the masked portion (i.e. the part of the trajectory given as input) supervised during training?

The classifier-free guidance (CFG) formulation was also a bit confusing. The difference between $x_{obs}$ and $x_{cond}$ in Sec 5.1 was not apparent until later in Sec 6.2. Usually CFG operates by dropping the entire conditioning from the model in the right-hand term of Eqn 6, so I’m wondering why only some of the conditioning (e.g. the goal state in Sec 6.2) is arbitrarily dropped in the proposed formulation? In this example (and in others like motion forecasting), it seems the past conditioning should also be dropped. I also think on L275-276 it may be more accurate to say that the CFG controls the emphasis of the conditioning to the model instead of controlling the “spread of agents’ trajectories”: the reason the spread reduces may be because the goal position conditioning is emphasized so more samples are guided closer to this goal.

The paper and supplement currently have only a few qualitative results, making it hard to comprehensively evaluate trajectory quality. Given the range of applications that the masking enables, it would be great to show several examples from each of these applications. Videos would be ideal, but at least trajectories should be colored with a gradient over time or something rather than a solid color over the whole trajectory.

The observation distribution is detailed in Appendix B, but I’m curious about how this distribution was determined and how much it affects the final capabilities of the model? It seems balancing these probabilities would be very important to ensure good results, so I’m wondering does the model do significantly worse when trained on all tasks jointly than trained on one individually?

One of the most useful applications of the proposed method could be in simulating scenarios for AV testing, but the current generated trajectories are quite short (at most 5 sec) and limits the kind of scenarios. I’m interested to see if the model can handle longer generated scenarios (~10 sec), and if the trajectories maintain realism using metrics like off-road and collision rate.

**Questions:**

Overall, the paper is a nice exploration and demonstration of how diffusion can be applied to traffic scenario modeling, and I tend to think it would be a good contribution to the community. But in the rebuttal, it would be really good to see more qualitative results and for the authors to clear up some of my confusion on the masking and classifier-free guidance procedures.

I had a few questions in Sec 4 that didn’t weight heavily in my rating:
* Eqn 5: why is the loss on $x$ and not $x_0$? Is the entire trajectory supervised (observed + generated) or just the non-observed parts?
* L186: why are the agent states called “latent”? Are they already embeddings of some kind?
* It’s interesting that deterministic sampling is better quality than stochastic since this is not the case in other domains – any intuition here?

============== After Rebuttal ===================

After the rebuttal and discussion, I have decided to raise my score to Accept. In the rebuttal, the authors addressed most of my concerns, clarified various technical points, and added more qualitative results that demonstrate the variety of applications.

**Limitations:**

Limitations are discussed in Sec 7. It might be good to mention some limitations/failures specific to DJINN and not diffusion in general, e.g., mention that the model is not always realistically consistent with conditioning observations?

---

> ### Author Rebuttal · Authors · 2023-08-09
>
> Consistency with input observations: We agree that in some examples, DJINN produces samples which are inconsistent across time. We also agree that these inconsistencies may be reduced by utilizing test time guidance. Specifically, classifier-free guidance as outlined in section 5.1 and demonstrated in section 6.2 might be used to improve the quality of these samples. In addition, iterative classifier guidance proposed by CTG is directly compatible with the classifier guidance outlined in section 5.2 and is a promising avenue for future work. We propose including a supplementary figure showing the effect of classifier-free guidance over a variety of guidance weights on the samples from Figure 1 in the final draft. While “Trace and Pace” [Rempe et al., CVPR 2023] is concurrent work by the NeurIPS standards and focuses on pedestrians instead of vehicle motion modeling, we believe it adds additional context for the reader and therefore will add it to our related work section.
>
> **Masking Procedure**: DJINN only produces predictions for agent states which are not observed. In Figures 1 & 2, we visualize the generated, unobserved states in orange and the observed states, which are conditional input to the model, in blue.
>
> **Classifier Free Guidance Formalism**: Thank you for identifying that the distinction between $x_{obs}$ and $x_{cond}$ in section 5.1 is unclear. We will modify the introduction of $x_{cond}$ to include a statement indicating that $x_{cond}$ could be a future goal state to improve the clarity.
>
> **Additional Conditional inputs in CFG**: Regarding your question about our classifier free guidance formulation, we agree that in many applications of classifier-free guidance (for example class conditioned image generation) a completely unconditional score estimate is combined with the conditional score estimate. From [13]:
>
> $ \nabla \log p(\mathbf{x}_t | y) =  (1-w) \nabla \log p(\mathbf{x}_t) - w \nabla \log p(\mathbf{x}_t | y)$
>
> Where x is the diffusion variable, y are conditional inputs and w is the guidance weight. However, it is also valid to add additional conditioning to every term:
>
> $ \nabla \log p(\mathbf{x}_t | y, z) =  (1-w) \nabla \log p(\mathbf{x}_t | z) - w \nabla \log p(\mathbf{x}_t | y, z)$
>
> In section 5.1, we define $y$ as $\mathbf{x}_{cond}$. Here, $z$ is all the other conditional information provided to the model, which in our application includes the map as well as other observed agent states. Our removal of specific agent states during guidance is therefore not arbitrary but instead an intentional design choice which follows the conventions of classifier-free guidance with added conditional information.
>
> **Phrasing of guidance strength**: We agree with the reviewer’s suggestion about the phrasing of line 275-276 and will update the final manuscript with this modification.
>
> **Additional Qualitative Results**: We have supplied some additional composite images showing scenario samples using a variety of observation masks. We agree that gradients on the qualitative result figures might improve clarity, and will update our figures with this modification.
>
> **Impact of the Observation Distribution**: The mixture weights for the various tasks in the observation distribution were chosen empirically. However, we have included an additional experiment in our rebuttal investigating this choice.
>
> | Model                	| Ego minADE | Ego minFDE | Scene minADE | Scene minFDE | Mean MFD |
> |--------------------------|-----------:|-----------:|-------------:|-------------:|---------:|
> | Predictive Only      	|   	0.21 |   	0.49 |     	0.35 |     	0.91 | 	2.33 |
> | Observation Distribution |   	0.26 |   	0.63 |     	0.45 |     	1.17 | 	3.11 |
>
> The table above shows the difference in performance on the INTERACTION dataset between a model trained on only the predictive task and one trained on the full observation mask distribution as measured using  6 trajectory random samples. As expected, we find the model which is optimized solely on the predictive task performs better on that task than the model which is trained on a wider distribution of tasks. We expect that scaling the network size of DJINN would likely reduce this difference.
>
> **Scaling**: We have only trained our model on datasets with scenarios of up to 5 seconds in length, as this is what is commonly used in challenges related to these datasets. We agree that generating longer scenarios is an important avenue of future work, but it is beyond the scope of this work. Therefore, in the final draft of our paper, we will discuss increasing the length of generated scenarios as an avenue of future research.
>
> **Eqn 5**: The loss we optimize is squared L2 error between the output of D_\theta and the noiseless unobserved states in the scene. We agree that the notation of equation 5 could be improved. In the final version of our paper, we will clarify the notation used in this equation.
>
> **L186**: We use the term “latent” as a synonym for “unobserved”, following the graphical model convention in which variables can be either latent or observed. We will update the text to improve clarity surrounding this point.
>
> **Deterministic vs Stochastic Sampling**: We note that EDM [21] also found that deterministic sampling performed better than stochastic sampling in their experiments. Since we utilize their diffusion parameterization, we are not surprised that deterministic sampling performs better in our experiments also.

---

> > ### Comment · Reviewer_wJNr · 2023-08-17
> > **Response to rebuttal**
> >
> > I thank the authors for their detailed response to my questions and concerns. Most of my issues have been resolved or clarified, and I intend to keep my rating of accept.
> >
> > However, for additional qualitative results I do not see an attached pdf to the individual response or the top-level response. Is it still possible to upload this?

---

> > > ### Author Response · Authors · 2023-08-17
> > > **Response to reviewer wJNr**
> > >
> > > Hello reviewer wJNr,
> > >
> > >  We are puzzled by the apparent disappearance of the additional qualitative results from our top-level response. When we read your rebuttal, we were also unable to locate the attached pdf. However, now we are able to see the qualitative results pdf again.
> > >
> > > It seems like there was some sort of issue with OpenReview where the pdfs were not visible. We apologize for the inconvenience. If you still are unable to see the qualitative response pdf please let us know and we will send a message to the AC/PC.

---

> > > > ### Comment · Reviewer_wJNr · 2023-08-17
> > > > **Qualitative results**
> > > >
> > > > I can see the attached pdf now! That is weird indeed. Great, thank you.

---

### Official Review · Reviewer_1Rcd · 2023-07-06

**Soundness:** 3 good
**Presentation:** 4 excellent
**Contribution:** 3 good
**Rating:** 6
**Confidence:** 4

**Summary:**

The paper proposes a generative model for producing synthetic traffic scenarios. The proposed method uses a diffusion model and learns to predict multiple agent trajectories jointly in a scene. The main contribution to previous methods is the increased flexibility of conditioning which can use different observation masks (corresponding to different use-cases), goal/non-goal, and other vehicle properties. The method is evaluated on two publicly available datasets and shows competitive performance to its deterministic counterparts.

**Strengths:**

DJINN seems like a natural step forward for the multi-agent trajectory prediction task. In theory, the generative framework of diffusion models should allow for better sampling of the data distribution and produce traffic scenarios with larger variability that previous methods such as Scene Transformer [30] and TrafficSim [44]. This is important because datasets with traffic scenarios are highly imbalanced, containing very few safety critical events which are usually more interesting. The paper illustrates this flexibility through several interesting qualitative examples of test-time conditioning. Furthermore, the paper is well written, providing clear motivation, problem formulation, and background to the reader.

**Weaknesses:**

Quantitative results in Tables 1,2 show comparable results to [30] in terms of forecasting performance but there is no quantitative experiment demonstrating the increased variability of generated traffic scenarios from the proposed approach. While this is shown qualitatively in sections 6.2, 6.3, and 6.4, and in my understanding [30] does not have the capability of test-time conditioning, I think a quantitative comparison (perhaps estimating variance over waypoint predictions) would still be useful. I would like to hear the authors' thoughts on this.

The authors mention that the choice of the reference frame for the agent states is important. Perhaps an ablation study would provide more insight.

What is the effect of using the observation masks during training besides defining different use-cases? How would the capability of the model to produce traffic scenarios with large variability be affected if only the "Predictive" mask was used?

Please include a runtime comparison to baseline methods.

**Questions:**

See weaknesses.

**Limitations:**

The paper did not address the limitations and potential negative societal impact.

---

> ### Author Rebuttal · Authors · 2023-08-09
>
> We thank the reviewer for taking the time to consider our work, and for providing positive, and constructive feedback. We address their concerns below:
>
> **Demonstration of increased variability**: We agree that such a comparison should be considered in the evaluation of our model in comparison to the baseline. One possible metric we could consider is the maximum final discrepancy (MFD) [38], which we have used in the “Effect of observation mask” subsection of this rebuttal to measure the variation of generated trajectories. If the reviewer believes that the inclusion of this evaluation would improve the clarity and impact of our work, it can be included in the final version of the paper.
>
> **Test-time conditioning**: We agree with the reviewer that surfacing rare events such as safety critical incidents is challenging with imbalanced driving data. The test time conditioning introduced in DJINN is one answer to this problem. Specifically, using classifiers of rare driving behavior and editing scenarios are two approaches introduced which allow sampling from conditional distributions of interest which are not available in prior methods.
>
> **Reference Frame**: Although the choice of reference frame is a key design choice, it is not easily modified without significant changes to the model architecture. This is because DJINN is specifically designed for a joint global frame, making extensions to an egocentric frame difficult . Therefore, we argue that such ego-centric prediction is outside the scope of this work, though we agree that it is a promising avenue for future research.
>
> **Effect of observation masks**: As the reviewer mentions, the primary reason for training over different observation masks is to define different use-cases at test time. To measure the impact of the observation mask distribution, we have compared the diversity of samples from a model trained with the “predictive” mask to those produced by a model trained on the full observation mask distribution on the INTERACTION validation set. We utilize the maximum final discrepancy as our diversity metric [38] which measures the maximum distance between all pairs of trajectories in a trajectory set.
>
> | Model                	| Mean MFD |
> |--------------------------|---------:|
> | Predictive Only      	| 	2.33 |
> | Observation Distribution |3.11 |
>
> From our experiment, we note that training over the observation mask distribution increases the diversity of trajectories produced by DJINN. However, we reiterate that the purpose of the observation distribution is primarily to enable test time guidance using classifier-free guidance and conditional generation of trajectories using arbitrary observation masks.
>
> **Runtime**: We have attached a comparison of the runtime of DJINN with [30] which will be added to the final supplementary. We note that the runtime can be lowered at the expense of performance as demonstrated in the rebuttal to reviewer 1Qe5, and that distillation is a promising avenue of future work to lower the computational cost of diffusion models.
>
> | Number of Agents | Scene Transformer | DJINN - 25 Steps | DJINN - 50 Steps |
> |-----------------:|------------------:|-----------------:|-----------------:|
> |            	8 |       	0.0126s |       	0.574s |       	1.149s |
> |           	16 |        	0.014s |       	0.611s |       	1.238s |
> |           	32 |        	0.017s |       	0.844s |       	1.693s |
> |           	64 |        	0.026s |       	1.404s |       	2.891s |

---

> > ### Comment · Reviewer_1Rcd · 2023-08-13
> >
> > I would like to thank the authors for their comprehensive answers to my comments.
> > I understand that the observation distribution is primarily for test time guidance, but I would suggest including the MFD results as they demonstrate 1) that your model does not collapse to a mean solution, and 2) that training over the observation mask distribution does indeed produce paths with larger variability which is desirable in this task.
> > After reading the other reviews and author rebuttals I am keen on keeping my original recommendation of acceptance.

---

### Official Review · Reviewer_TiFk · 2023-07-06

**Soundness:** 4 excellent
**Presentation:** 4 excellent
**Contribution:** 2 fair
**Rating:** 6
**Confidence:** 3

**Summary:**

The paper proposes a diffusion model-based method for generating joint or conditioned predictions of traffic agents. The proposed method combines the EDM diffusion and SceneTransformer model architecture and provides different guidance for conditional predictions. The experiment results indicate comparable results of the proposed results with SOTA.

**Strengths:**

The paper is well-written and the proposed approach gives readers very useful information about how the two SOTA methods (EDM and SceneTransformer) could be combined and its potential performance.

**Weaknesses:**

In my understanding, the novelty of the paper is mostly on the combination of existing methods, which is OK but not outstanding.

From the experiment results, the proposed method doesn't show clear improvements compared with other methods, which is a bit surprising given the combination of two powerful approaches. It would be great if the author could give more analysis and explanations on why this is the case.

**Questions:**

It would be great if the author could give more analysis and explanations on why the performance of the proposed method doesn't show clear improvements.

**Limitations:**

As mentioned in the paper, the long inference time limits the method to be more useful in practice.

---

> ### Author Rebuttal · Authors · 2023-08-09
>
> Thank you to reviewer TiFK for their thoughtful review of our work and their praise of our paper’s clarity. Below, we have responded to the two areas of weakness which the reviewer has highlighted.
>
> **Novelty**: In regards to the novelty of our approach, we accept that our method combines the training and inference structure of EDM [21], while using a model backbone similar to Scene Transformer [30]. However we emphasize that in combining these two approaches we enable capabilities which are not present independently in either work. DJINN allows increased control over traffic scene generation through guidance and scenario editing. Specifically, sampling traffic scenes using arbitrary trajectory classes and editing scenarios are both capabilities which are not available in either work.
>
> **Performance**: We argue that the main contribution of our approach is not the predictive power of DJINN as a trajectory forecasting model, but the ability to sample traffic scenes from a wide variety of conditional distributions through guidance and editing techniques as outlined in section 5 and demonstrated in experiment sections 6.2-6.4.
>
> Regarding raw performance, we first highlight that DJINN is constructed to perform joint trajectory prediction and not marginal trajectory prediction. In joint trajectory forecasting, DJINN outperforms a comparable baseline as is shown in Table 2, section 6.1.
>
> When comparing our joint model against marginal methods in Table 1, we highlight that the standard trajectory forecasting metrics, minFDE and minADE, measure the minimum error over a small set of predicted trajectories. We argue that these metrics implicitly make a strong modeling assumption that the true distribution of driving behavior follows a finite component mixture distribution.
>
> The non-generative trajectory forecasting baselines considered in our work incorporate this assumption by directly modeling discrete, deterministic trajectory sets. These sets contain the average driving behavior of each component in the mixture, but might not capture the true driving distribution. Alternatively, DJINN is trained to stochastically produce traffic scenarios which match the data distribution by minimizing a weighted variational bound with weaker assumptions about the structure of the underlying data distribution.
>
> We argue that the slightly worse performance of DJINN relative to these models stems from the shared implicit assumption made by common evaluation metrics like minFDE and minADE and the baseline models we compare against. Utilizing the post-processing described in section 6.1 to approximate DJINN’s learned distribution with a six component Gaussian mixture results in reasonable forecasting performance under these metrics. However, we hypothesize that more direct minimization of these metrics through the shared mixture assumption described above is the primary reason for the performance discrepancy.

---

### Official Review · Reviewer_1Qe5 · 2023-07-11

**Soundness:** 4 excellent
**Presentation:** 3 good
**Contribution:** 3 good
**Rating:** 8
**Confidence:** 3

**Summary:**

DJINN (Diffusion-based Joint Interaction Network) is an innovative generative model that creates, edits, and forecasts multi-vehicle traffic scenarios in a stochastic manner. Leveraging diffusion models, DJINN also addresses the challenge of generating traffic scenes conditioned on a flexible configuration for the observation space. This flexibility of the model makes it well-suited for generating safety-critical events and out-of-distribution scenarios, which are extremely relevant for evaluating the performance of autonomous vehicles. Furthermore, by jointly diffusing the trajectories of all agents and conditioning them to customizable observation windows, the authors provide a fresh and innovative way of forecasting multi-vehicle traffic scenarios.

The authors validate the efficacy of DJINN by benchmarking against state-of-the-art trajectory forecasting methods using the popular Argoverse and INTERACTION datasets. The results show that DJINN outperforms existing models like Scene Transformer in joint motion forecasting metrics, demonstrating its promise for contributing significantly to the development and safety testing of autonomous vehicles. Moreover, DJINN’s flexibility is demonstrated through its successful generation of goal-directed samples, examples of cut-in driving behaviors, and editing replay logs.


**Strengths:**

Originality:
The paper represents a highly original contribution to the field of autonomous vehicle simulation. The use of a diffusion model for generating joint traffic scenarios is innovative, and appears to be a novel application of this technique. The authors move beyond the conventional deterministic sets of trajectory forecasts, proposing a generative model to forecast joint future motion. DJINN's ability to draw traffic scenarios from a variety of conditional distributions further attests to its innovative design. The model also stands out in its ability to provide flexibility in terms of test-time diffusion guidance, offering a new perspective on traffic scenario simulation.

Quality:
The overall quality of the research is high. The paper presents a sound background and solid related works section, creating a strong foundation for their argument. The authors have clearly demonstrated their understanding of the topic, using relevant and current research. Their methodology is rigorous, and the experiments performed on the Argoverse and INTERACTION datasets are thorough and well-executed. The results obtained are credible and validate the authors' claims about DJINN's performance and flexibility.

Clarity:
The paper is well-written and clear. It efficiently communicates the problems associated with the simulation of autonomous vehicle systems and convincingly presents DJINN as a potential solution. The structure of the paper is logical, and the flow of ideas is coherent, making it easy for readers to follow the authors' argument.

Significance:
The significance of this paper is recognizable. Simulating diverse, safety-critical and stochastic traffic scenarios is of paramount importance in the field of autonomous vehicles.  Its flexibility in controlling the conditioning of traffic scenes, coupled with its excellent performance on trajectory forecasting, makes it a valuable tool for researchers and developers in the field. Given the increasing interest in autonomous vehicles, this work is likely to have a high impact on both academia and industry.

**Weaknesses:**

While the paper represents a significant contribution to the field, there are a few areas where the authors could improve upon in the future iterations of their work:

- Detailed Analysis of Limitations: While the conclusion does mention certain limitations, the paper would benefit from a more detailed and separate section dedicated to the limitations of the proposed method. A thorough examination of the constraints and failure cases of DJINN could offer readers a more balanced perspective and help to guide future research efforts.

- Future Directions: The authors might also consider providing clearer guidance on the future direction of the work. While DJINN has shown promise in its current state, discussing potential extensions and enhancements would be valuable. For instance, how might DJINN be adapted to handle more complex scenarios, or different action spaces and robotic domain?

- Qualitative Analysis of Failure Cases: Alongside quantitative performance metrics, a qualitative analysis of failure cases would provide more comprehensive insights into the model's performance. Discussing specific scenarios where DJINN fails to generate accurate traffic scenes would offer readers a better understanding of its potential weaknesses and areas for improvement.

- More Sample Traffic Scenes: The paper could also benefit from a larger set of sample traffic scenes, either included in an appendix or made available online. These examples would provide a more tangible sense of the model's capabilities. Video demonstrations would be particularly effective, allowing readers to visually appreciate the flexibility and realism of the scenes generated by DJINN.

- Ablation Studies: The paper could also greatly benefit from conducting ablation studies to understand the influence of different components or variations of DJINN's model architecture on its overall performance. A detailed ablation analysis can help to pinpoint which elements of the model contribute most to its successful generation of traffic scenarios. This could include investigations into the impact of the diffusion process, the role of different conditional distributions, or the effect of different types of state observations. Understanding these aspects in more detail would not only provide further insights into DJINN's performance and functioning, but also guide future optimization and refinement of the model.

Minor comments:
Unlinked reference at line 88.


**Questions:**

- How might DJINN be adapted to handle more complex scenarios, higher action spaces or other robotic domain?
- Given that the inference time of DJINN it relatively slow, due to its diffusion structure and iterative estimation of the score function. Do you have any quantitative results on the tradeoff between computing time, performance and variation of network? (number of layer, depth, number of diffusion steps, etc)
- How suitable would the model, as it is, be for a model of predictive control setting and real life deployment?

**Limitations:**

While the authors acknowledge some limitations of DJINN in the conclusion, particularly the relatively slow inference time, a more comprehensive and explicit discussion of these drawbacks would enhance the overall balance and depth of the paper.

Moreover, this section could provide a more explicit roadmap for future research directions. While the paper showcases DJINN's potential, there is room to explore its applicability in other contexts, or the implications of further improving its inference speed. Articulating these directions can inspire and guide subsequent work in this line of research.

---

> ### Author Rebuttal · Authors · 2023-08-09
>
> We would like to thank reviewer 1Qe5 for taking the time to consider our work, as well as for their overwhelmingly positive and constructive feedback. We consider the areas which the reviewer believes that our work can be further improved below:
>
> **Detailed Analysis of Limitations**: We agree that additional discussion of the limitations could improve the clarity and impact of our work. In order to address the reviewer's comment, we plan to include an additional ablation study in the final draft of the paper which we feel will provide the reader with a more intuitive understanding of our model and its limitations. Specifically we will look at how DJINN’s run-time scales under increased agents and scenario lengths. This is a well known shortcoming of transformer models, which by extension also affects DJINN itself.
>
> **Future Directions**: We note that a small discussion of future research areas is included in section 7 (line 321), however in the final draft we will further elaborate on these directions, including many of the useful suggestions provided by the reviewers.
>
> **Qualitative Analysis of Failure Cases**: We agree that consideration of failure cases can often provide insight into what the limitations of the model might be in practice. However during our experiments we found that failure cases were most often the result of poor dataset annotations, and not any fundamental underlying problem with our modeling approach.
>
> **More Sample Traffic Scenes**: We thank the reviewer for this suggestion, which we agree would enhance the quality of our work. We have attached a pdf containing an additional figure in our rebuttal showing additional qualitative results for our method. This figure provides samples using the predictive, goal conditioned, agent reactive, and up-sampling tasks outlined in Figure 1 of the main text. If accepted, we will add this figure to the supplementary. Additionally, in the final draft we can include a link to a paper web-page including video examples of our method.
>
> **Ablation Studies**: In order to address the reviewer’s concerns, we have included the following ablation result regarding the effect of varying number of diffusion steps on the quality of generated traffic scenarios. We evaluate our INTERACTION model using a variety of diffusion steps, evaluating metrics over a 6 trajectory samples per scene, and refrain from fitting a Gaussian mixture model.
>
> | Diffusion Steps | Ego minADE | Ego minFDE | Scene minADE | Scene minFDE |
> |----------------:|-----------:|-----------:|-------------:|-------------:|
> |              10 |       0.28 |       0.64 |         0.45 |        1.135 |
> |              20 |       0.22 |       0.51 |         0.37 |         0.95 |
> |              30 |       0.22 |       0.50 |         0.36 |         0.92 |
> |              40 |       0.21 |       0.50 |         0.35 |         0.92 |
> |              50 |   **0.21** |   **0.49** |     **0.35** |     **0.91** |
>
>
> Reducing the number of diffusion steps decreases the predictive accuracy of the model, but improves the runtime, allowing the user to trade performance for speed. The other ablation which was requested in the questions section of the review regarding the varying of model layer depth will be included in the appendix of the final draft of the paper.
>
> **More complex scenarios**: We note that because DJINN is based on a transformer model, we expect that with larger datasets, the model's performance would continue to scale desirably. This property has been well studied in other transformer networks [Kaplan and Mcandlish, 2020]. While adapting DJINN to more complex robotics domains represents an interesting future direction, it is beyond the scope of this work. DJINN is tailored specifically to autonomous driving, and while the diffusion framework can be adapted to arbitrary state spaces, determining how best to learn in such state spaces would require additional consideration. We will add this as a suggestion for future work into our final manuscript.
>
> Computing time, performance and variation of network: As is discussed in Ablation Studies, we have included an experiment which considers the effects of varying the number of diffusion steps. In our response to reviewer 1Rcd, we have also measured the runtime of our method over varying numbers of agents and diffusion steps.  An experiment which varies the network structure will be included in the final draft of the paper.
>
> **Suitability in MPC**: DJINN could be adapted to a multi-agent, model predictive control setting (MPC), where a local dynamics model is known but the external agents’ models are unknown. MPC could proceed by taking an action, stepping the local dynamics model to determine the next egocentric state, then using DJINN to predict the states of all external agents in the scene. Such a method could function as a basic MPC algorithm, or could even be used for interactive simulation and driving policy evaluation. Additionally, we hypothesize that MPC might be useful in reducing collisions in generated scenarios, but evaluation of this hypothesis is beyond the scope of our current work.
>
> We agree that investigation into how generative models like DJINN might be applied to real-world deployment represents an important research direction as autonomous vehicle technologies continue to evolve. However, considering the difficult research questions surrounding closing the sim2real gap and diffusion model runtime, DJINN would be unlikely to be suitable for onboard, real-time deployment. That said, we will add a discussion of these research questions to our final manuscript.

---

### Author Rebuttal · Authors · 2023-08-09

We would like to thank all of the reviewers for their time in considering our work and their thoughtful comments which we believe will help to improve the quality of our submission.

In response to requests for additional qualitative results, we have provided an attached pdf which contains a composite image of multiple traffic scenes generated by DJINN under a variety of observation masks.

---

### Decision · Program_Chairs · 2023-09-21

**Decision:**

Accept (poster)

**Comment:**

The paper proposes a diffusion model for generating joint traffic scenarios. The reviewers appreciate the clear motivation, good presentation and comprehensive ablations despide of a limited novelty on components of the method. All the reviewers agreed to accept the paper after the rebuttal. Therefore, the AC recommends accepting the paper.